# Use of Tyre-Derived Aggregate as Backfill Material for Wave Barriers to Mitigate Railway-Induced Ground Vibrations

**DOI:** 10.3390/ijerph17249191

**Published:** 2020-12-09

**Authors:** Jesús Fernández-Ruiz, Luis E. Medina Rodríguez, Pedro Alves Costa

**Affiliations:** 1Department of Civil Engineering, University of La Coruña, 15071 La Coruña, Spain; luis.medina@udc.es; 2Department of Civil Engineering, University of Porto, 4200-465 Porto, Portugal; pacosta@fe.up.pt

**Keywords:** tyre-derived aggregate (TDA), railway vibrations, pile wave barriers, mitigation measures

## Abstract

The use of piles as barriers to mitigate vibrations from rail traffic has been increasing in theoretical and practical engineering during the last years. Tyre-derived aggregate (TDA) is a recycled material with some interesting applications in civil engineering, including those related to railway engineering. As a novelty, this paper combines the concept of pile wave barriers and TDA material and investigates the mitigation effect of pile barriers made of TDA on the vibrations transmitted by rail traffic. This solution has a dual purpose: the reduction of railway vibrations and the recycling of a highly polluting material. The mitigation potential of this material when used as backfill for piles is analysed using a numerical scheme based on a 3D finite-difference numerical model formulated in the space/time domain, which is also experimentally validated in this paper in a real case without pile barriers. The numerical results show insertion loss (IL) values of up to 11 dB for a depth closed to the wavelength of Rayleigh wave. Finally, this solution is compared with more common backfills, such as concrete and steel tubular piles, showing that the TDA pile is a less effective measure although from an environmental and engineering point of view it is a very competitive solution.

## 1. Introduction

The development and application of numerical models for the study and prediction of railway vibrations has progressed significantly in recent years with outstanding models such as those presented in [1,2,3,4,5,6,7], among others. The study of mitigation measures has also received significant attention among researchers due to the negative impact of railways in terms of noise and vibrations, especially appreciable in urban environments. It is well known that these measures are applicable in three different areas: (i) in the rail tracks, with elastic mats used under the track [8], subgrade stiffening [9], and improvements in rail irregularities and wheel defects, among others; (ii) in the wave propagation path, such as open trenches and filled trenches [10,11,12,13,14,15,16] among others and buried periodic inclusions [17,18,19,20,21,22,23,24,25,26,27,28,29,30,31]; and (iii) on the building, through base isolation solutions [32].

There are many studies on open- and filled-trench measures, but this paper highlights the study by Thompson et al. [10], which shows how open trenches are more effective measures than filled trenches, although open trenches can represent a safety problem. Moreover, filled trenches are also interesting solutions when these are backfilled with very soft materials since it causes a noticeable reduction in vibrations. In this regard, Thompson et al. [10] state that the key parameter for a significant reduction in vibrations in filled trenches is the stiffness contrast between the ground and the barrier material and not the impedance, as occurs in cases that consider the transmission at the interface between two semi-infinite mediums. In this regard as well, Barbosa et al. [12] have shown that highly stiff backfills (such as concrete) cause greater reduction in the level of vibrations than backfills with less stiff materials, such as geofoam. In the cases of stiff materials (stiffer barriers), the reduction effect is mainly induced by the guided wave effect caused by surface waves coupling and propagating in the ground and the bending waves propagating in the barrier.

Large quantities of tyre-derived material are generated annually worldwide. Recent studies show production of 6.4 and 12.6 kg/resident/year in Europe and the USA, respectively [33]. Logically, when this material is no longer usable due to deterioration, it becomes a waste and, given the environmental risk involved, many efforts have been made to increase its recycling and/or reuse. A clear example is its recycling for use in bitumen for the manufacture of bituminous mixtures. However, a simpler and more efficient recycling process in terms of energy consumption is tyre-derived aggregate (TDA) [34], which is obtained from cutting scrap tyres into relatively large pieces (25–300 mm). TDA has interesting geotechnical properties such as vibration damping potential [35] and high permeability [36], among others. With regard to railway vibration mitigation measures, it seems reasonable that its high damping may be an interesting property that combined with its low Young’s modulus value makes it, a priori, a suitable material for filled trenches or as backfill for piles. Its applications in civil engineering are continuously increasing, and its road and railway engineering applications stand out [37,38,39,40,41,42,43]. In addition, it has been used in easing embankments on soft soils and on tunnels, the latter built with the cut-and-cover system [33]. However, it has not been specifically used as a material for filled trenches or pile wave barriers as a mitigation measure for rail traffic vibrations.

For this reason, this paper studies for the first time the efficiency of pile barriers made exclusively of TDA in the reduction of railway vibrations. In this regard and given the characteristics of this material, in practice, its use in barriers is only possible for tangent or spaced piles with a recoverable steel tubular pile. Its implementation with mediums similar to the usual concrete diaphragm walls is not possible due to the lack of lateral confinement between the adjoining panels. Its practical use is possible in trenches but requires moderate trench depths (<4 m); therefore, these are excluded in this paper since it considers barriers with greater depths, because they are more effective for vibration reduction than shallow barriers.

As for the organization of this paper, the following methodology was followed: (i) first, the numerical model used, which is based on a three-dimensional finite-difference model formulated in the time domain, is described; (ii) the numerical model is experimentally validated with real measurements from passing trains, without mitigation measures; (iii) the effect of the depth of the barrier and the effect of the spacing between pile axes caused by a Ricker-wavelet-type stationary point load is studied; (iv) once the depth and axis spacing conclusions are obtained, the level of vibrations in the ground from the passing train is compared with that when the TDA pile barrier is introduced; (v) the influence of the constitutive model for the TDA material is analysed and the linear elastic model is compared with a hyperbolic model and with an anisotropic model; and lastly (vi) the vibration reduction of the TDA barrier is compared with that of barriers made of other more usual backfills, such as concrete and steel tubular piles.

Therefore, this paper studies, for the first time, wave pile barriers made exclusively with TDA as a vibration mitigation measure for rail transportation systems. This is novel from an environmental and engineering point of view because a highly polluting waste can be reused as a backfill material for the reduction of railway vibrations. Hence, the main aims of this study are to analyse the level of vibration reduction on the ground caused by this kind of pile barrier, studying both the effect of the depth and the pile spacing.

## 2. Materials and Methods

### 2.1. Generalities

As a preliminary phase to the study of the effect of TDA pile barriers, the numerical model without mitigation measures was experimentally validated in a real case in Portugal. This paper uses a sub-structured numerical approach as follows: (i) the dynamic train–track–ground interaction is solved by using a 2D model and the results are used to estimate the transmitted load to each sleeper; (ii) the latter is introduced into a numerical 3D finite-difference model formulated in the space/time domain on FLAC software, where the propagation of waves and ground response is studied. The computational scheme is summarized in Figure 1.

### 2.2. Train–Track–Ground Interaction

The train–track–ground interaction model has been considered as shown in Figure 2, where all masses of the train have been taken into account. In this model, the train is completely considered (including all the masses) through the rigid body dynamics method with 10 degrees of freedom. The track is simulated as a combination of masses and linear spring–dashpots, and the ballast, sub-ballast, and ground are replaced by a linear equivalent spring–dashpot. The rail and the sleepers are modelled as infinitive Euler–Bernoulli beams and the rail pad as a linear spring–dashpot. The train and track are interrelated through the wheel–rail Hertz contact. The resultant forces over the sleepers are computed and later applied in the numerical model, as summarized in Figure 1. Similar models are common in the technical literature [44] and elsewhere.

It must be highlighted that only vertical movements have been taken into account because these are the main source of railway vibrations. Additionally, the sole dynamic excitation mechanism considered was the vertical irregularities of the rail. A detailed description of the mathematical formulation inherent to this interaction model can be found in [1]. It should be pointed out that these types of 2D models have limitations, such as not considering the discrete nature of the sleepers. However, the application and suitability of these models have been extensively tested by many authors [1,6,45,46,47] and are considered sufficiently valid to be used in this research.

Once the dynamic train–track–ground interaction has been solved, the force on each sleeper F_s,i_ (t) is obtained with the following expression:(1)$$F_{s,i}\left(t\right)=\left(u_{\mathrm{rail},i}-u_{\mathrm{sleeper},i}\right)K_{\mathrm{rp}}\;$$
where u_rail,i_ and u_sleeper,i_ are the displacements of the track and the sleeper in i position, respectively, and K_rp_ is the vertical stiffness of the rail pad.

### 2.3. Explicit 3D Finite-Difference Method Formulated in the Space/Time Domain

Given the geometric characteristics of the pile barriers analysed in this paper, a detailed and accurate 3D numerical approach is necessary. Therefore, this numerical model has been developed with the FLAC3D software. This is an explicit finite-difference formulated in the space/time domain to analyse the mechanical behaviour of a continuous 3D medium. The fundamental equations of its mathematical formulation can be consulted in [48]. The basic equation of dynamic equilibrium is written in the following form:(2)$$\sigma_{\mathrm{ij},j}+{\rho b}_{i}=\rho \frac{{\mathrm{dv}}_{i}}{\mathrm{dt}}$$
where ρ is the mass-per-unit volume of the medium, bi is the body force per unit mass, and dv_i_/dt is the material derivate of the velocity.

Special attention must be paid to the critical time step in an explicit finite-difference approach with a stiffness-proportional damping. In this case, the critical time step (Δt_crit_) is formulated as follows [48]:(3)$$\Delta t_{\mathrm{crit}}=\left\{\frac{2}{\omega_{\max}}\right\}\left(\sqrt{1+\lambda^{2}}-\lambda \right)$$
where ω_max_ is the highest eigen frequency of the system, and λ is the fraction of critical damping at this frequency. They are given by the following expressions:(4)$$\omega_{\max}=\frac{2}{{\Delta t}_{d}};\;{\Delta t}_{d}=\min\left\{\frac{V}{C_{p}A_{\max}^{f}}\right\}/2;\;\lambda =\frac{0.4\beta }{{\Delta t}_{d}};\;\beta =\frac{\xi_{\min}}{\omega_{\min}};$$

C_p_ is the p-wave velocity, V is the tetrahedral sub-zone volume, and A^f^_max_ is the maximum face area associated with the tetrahedral sub-zones ξ_min_ and ω_min_ are, respectively, the damping fraction and the angular frequency specified for Rayleigh damping.

The quiet (absorbing) boundaries have been considered, according to the formulation proposed in [48] and [49], and a Rayleigh-type damping has been adopted for the soils. These types of dynamic boundaries and soil damping are very common in numerical models formulated in the space/time domain, and its use is widely extended. For this reason, its mathematical description is omitted in this paper, and the reader is advised to consult [48].

### 2.4. Study Case

The numerical scheme used was experimentally validated on the Lisbon–Porto line, located near the town of Carregado (Portugal) by comparison between vertical vibrations at different points on the ground. These vibrations were induced by railway traffic in common traffic operations. The experimental campaign carried out at this location has been shown in several papers and consisted in measuring the following: (i) the geotechnical properties of the ground; (ii) the mechanical properties of the track (including ballast and sub-ballast); and (iii) the level of vibration induced by rail traffic in the track and in the ground. The results obtained and the details of this extensive experimental campaign can be consulted in [1,46], among others. This paper only shows the summarized properties considered for the ground (Table 1), for the track (Table 2), the profile of rail irregularities (Figure 3), and a scheme of the train (Figure 4). The Alfa Pendular train has been considered because the problem associated with railway-induced vibrations on the ground is usually limited to passenger trains (without heavy axle loads) where its high speed causes high-frequency vibrations that are perceived far away from the railway track (tens of metres). On the contrary, the highest vibrations produced by freight trains (heavy axle loads) are limited to the railway track, since its low speed implies low-frequency vibrations (<10 Hz) that are not perceived at important distances of the railway track.

The rail type is UIC-60 and the vertical stiffness of the rail pad is 600 kN/mm. The train speed was 164 km/h. More detailed information can be found in [1]. Related to the ground and track elastic properties shown in Table 1 and Table 2, it should be noted that all values were obtained from dynamic tests (cross-hole, receptance tests, etc.). As for the damping values, they have been considered as shown in [1,3].

The TDA material is made from used conventional commercial tyres. Table 3 shows its mechanical properties, considering a linear elastic model [33]. These properties correspond to the gradation of the TDA material shown in Table 4 and were obtained under laboratory conditions. However, there are some available field tests, and similar results were obtained [50]. Although there is still no experimental evidence in civil engineering applications, it is well known that the durability and lifetime of this material is very long, beyond the design life of the constructions in which it is used. The TDA material is impermeable and inert, and its mechanical behaviour depends very little on the environmental conditions where it is employed.

The shared node method is considered for the TDA piles–ground interaction because the relative movement between them is very small, so the friction between piles and soils is neglected in this research. In the same way, the interaction between different pieces of tyre rubber has also been neglected as the TDA is modelled as a continuous medium.

Figure 5 shows the numerical model used. It should be noted that, since the case is symmetrical, only half the problem was modelled. All contours correspond to absorbing boundaries, except in the plane of symmetry, where perpendicular displacements to it are not allowed and the ground surface corresponds to a free boundary.

For the experimental validation of the numerical model, the vertical vibration velocity values have been compared at points located on the central plane of the numerical model and at the following distances from the symmetry axis: 7, 15, 22.5, 30, 37.5, and 45 m. Figure 6 shows the results obtained in the time domain, where a reasonable match between computed and measured results can be observed. However, the agreement is not the same at all points, showing that at the points closest to the track the differences are smaller than those at the points located at distances greater than 30 m. These differences in the results in the time domain are common in railway vibrations and similar differences were found in [1,51], among others. From a global point of view, the results can be considered admissible.

Figure 7 shows a clearer comparison using the results in the frequency domain (one-third octave spectra).

As can be seen, the numerical results follow the same trend of the experimental data and, from a general point of view, an acceptable agreement between measured and computed was found. In general, the differences were not greater than 10 dB, which is an acceptable accuracy value, as shown in [47,52,53]. In some frequency bands, the differences were slightly greater than 10 dB, e.g., 20 Hz at the point located at 7 m and 16 Hz at the point located at 45 m. Even so, in general terms, the numerical scheme is considered adequate since it reproduces the experimental results with an acceptable level of accuracy. The accuracy of the numerical model in this paper is very similar to that in other papers in related literature ([1,45,47,51], among others).

## 3. Results and Discussion

### 3.1. Introduction

Before the efficiency of TDA-backfilled pile barriers under rail traffic was analysed, a preliminary study was carried out to examine the effect of pile depth and spacing. This study was carried out under a Ricker-wavelet-type stationary point load applied on the central sleeper of the numerical model formerly used. The use of stationary point loads is commonly applied to the study of wave-impeding barriers for the reduction of railway-induced ground vibrations, e.g., within a previous analysis to the study of the effects induced by moving loads, as can be seen in [9,13,17], among others.

In all the numerical models made in this paper, the diameter of the piles was 0.6 m, which may be considered as the smallest dimension in practice. A smaller diameter could be problematic during the construction and filling processes. Only this diameter has been considered in this paper because the barrier width seems not to have a relevant influence on results [10]. Moreover, the length of the pile barrier has been considered as 4, 8, and 12 m.

### 3.2. Vibration Due to Stationary Point Load

Figure 8 shows the stationary point load considered, where the frequency range studied is between 0 and 100 Hz.

To show the numerical results obtained, eight reference points were considered, in accordance with Figure 9. In addition, it is seen that the distance from the barrier to the axis of the track is 10 m, and this magnitude was constant for all the models carried out in this paper.

#### 3.2.1. Effect of Pile Length

Several studies on pile barriers or even filled trenches have shown that depth is a determining factor in the effectiveness of barriers for the reduction of railway vibration. Pu et al. [21] stated that the depth of the pile should be similar to the wavelength of the Rayleigh wave, for a given frequency, since a longer or shorter wavelength does not imply greater reduction. This should obviously occur, but it poses a problem at low frequencies since its associated wavelength is long and potentially causes piles with high lengths that may not be economically effective. Thompson et al. [10] analysed the effect of depth for open and filled trenches in a layered soil and presented results for three barriers of different depths. The numerical results show that starting from a given trench depth, the effect of lengthening the trench further may be negligible in terms of vibration reduction, depending on the characteristics of the layered soil and is clearly influenced by the propagation characteristics of Rayleigh waves in the ground. In that study, an increase in trench depth improved the reduction of vibrations only at medium and high frequencies. What is clear is that the mitigation of low frequency requires “unacceptable” lengths of trenches or piles, from an economic perspective. In addition, it must be considered that the low frequencies transmitted by rail traffic are usually attenuated in the vicinity of the track and the vibrations at points farther from the track are dominated by medium and high frequencies. In this paper, to study the effect of the depth of the TDA pile barrier, three depths were studied: 4, 8, and 12 m (Figure 10) and in all cases with tangent piles.

Figure 11 shows the comparative insertion loss (IL) results for the three depths at 15 m from the track axis. The IL is defined as follow:(5)$$\mathrm{IL}\;\left(\mathrm{dB}\right)=20\log_{10}\frac{\left|v_{\mathrm{ref}}\right|}{\left|v\right|}$$
where v_ref_ is the computed vertical velocity in the numerical model without pile barriers, and v is the computed vertical velocity in the numerical model with pile barriers.

As seen, the values for depths of 8 and 12 m are similar at the 4 control points. The differences between them are restricted to frequencies higher than 70 Hz while in low and medium frequencies the differences are not truly relevant. However, the differences with the 4 m long barrier are more important since there are differences of up to 10 dB for frequencies above 15 Hz. Below this value, the differences between the three lengths are negligible. Even though the ground is layered, the cut-off frequency can be related to the depth of the barrier. The S-wave velocities of the ground are between 140 and 225 m/s for the upper layer. For these wave velocities, an 8 m barrier is effective for frequencies above approximately 18 Hz, while the 12 m barrier is effective starting at 11 Hz, values that are shown approximately in Figure 11. In this context, it seems reasonable to think that an 8-m-long barrier is sufficient to combine economic efficiency and vibration reduction.

In similar terms and as Figure 12 shows, the aforementioned for the points at 15 m from the track axis can be extrapolated to the profile located at 30 m from the axis, where it is seen again that the reduction achieved by the 8 and 12 m long piles is similar, while the barrier with a depth of 4 m implies less vibration reduction and the differences with the other two cases are more significant. This shows that once the barrier penetrates the softer layers of ground, the effect of deepening the barrier into more stiff layers is limited and from a practical point of view, it is not very effective since the extra reduction provided is not significant. In this case, the stiff layer corresponds to soil layer 7, with E = 339 MPa and a depth between 6.5 and 12 m. Similar results are shown by [10] in a layered soil, where once the barrier reaches the stiff soil stratum, deeper depths into this stratum have a limited effect on vibration reduction. Furthermore, in that case, the differences found when the barrier is deepened further in the stiff stratum are limited to frequencies above 63 Hz and, in IL terms, barely reach 3 dB.

#### 3.2.2. Effect of Pile Spacing

The effect of pile spacing for the case of periodic wave barriers has been studied by several authors. Pu and Shi [19] showed how a greater distance between piles implies a smaller frequency band where vibrations are reduced. In addition, shorter distances between piles increase the reflected waves, thus improving the mitigation effect of pile barriers. Kattis et al. [18] compared the effectiveness of a pile barrier with a concrete trench of the same width and depth and found that pile spacing has a negative effect on the effectiveness to reduce vibrations.

In the case of barriers with TDA material, the effect of separating the piles was studied here, comparing the solution for tangent piles to a different arrangement where pile spacing is equal to 2Φ (Φ = pile diameter). Figure 13 shows the numerical model used with the two cases studied: tangent piles and spaced piles. In both cases, the depth of the barriers was 8 m. 

The results obtained are shown in Figure 14 (for a distance of 15 m to the track) and in Figure 15 (for a distance of 30 m to the track). As seen in the figures, the effect of spacing the piles 2Φ causes the barrier to have almost no mitigation, with maximum IL values of 2–4 dB and of up to 15 dB when the piles are tangential. If the two scenarios are compared, it is evident that spacing the piles with TDA material is not adequate for vibration reduction. This is because the isolated pile does not have vibration modes since its bending stiffness is very low. If the backfill material had a high Young’s modulus (for example concrete), the pile would have vibration modes and could be an effective measure to reduce vibrations, obviously to a greater or lesser extent depending on pile spacing. When piles are tangential, the mitigation effect is substantial due to the high damping capacity of the material and the stiffness contrast between pile material and the ground. From the results obtained, it is evident that discontinuous TDA structures are ineffective solutions for the reduction of vibrations.

### 3.3. Vibration Due to a Passing Train

Based on the results on the effect of pile depth and pile spacing obtained in the previous section, it seems clear that pile barriers with TDA must be tangential and their depth is closely related to the wavelength of the Rayleigh wave in the ground. Therefore, the mitigation effect of an 8-m-deep barrier of tangent TDA piles was analysed in the same case used in this paper to experimentally validate the numerical scheme.

Figure 16 compares the results in the time domain for different distances from the track axis. As seen, the pile barrier reduces the levels of vibration at the points closest to the barrier. Specifically, the maximum vertical speed peaks are reduced by half (at the points located at 15 and 22.5 m) and 30–40% at point located at 30 m, while at 45 m from the axis of the track, the effect is almost negligible, just as occurs with this type of mitigation measures.

A clearer analysis can be obtained from Figure 17, which shows the IL value caused by the barrier. It can be seen that the reduction reaches 10 dB in some frequency bands and that the mitigation decreases as the distance to the barrier increases. In particular, the IL values for points located closest to the barrier (15, 22.5, and 30 m from the track axis) are between 0 and 10 dB, and the frequency band between 15 and 60 Hz is the most damped. In addition, it is observed that for frequencies of up to 12 Hz, the mitigation effect is very low since wavelengths corresponding to lower frequencies are larger than the depth of the barrier. At the point located 45 m from the axis of the track, the mitigation effect of the barrier is negligible.

With these results, it can be stated that TDA pile barriers can be effective measures for the reduction of vibrations induced by rail traffic. In addition, the economic cost is very low, and it is an environmentally sustainable measure since a highly polluting material can be reused as an environmental measure, such as the reduction of railway vibrations. This kind of solution causes a reduction in vibrations because the damping of the TDA material is high and its stiffness is very low when compared to that of the ground. Moreover, the mitigation of this type of soft materials could be explained by a combination of diffraction under the barrier and transmission through the barrier itself [10].

### 3.4. Influence of the Constitutive Model for TDA Material

As shown above, the behaviour of the TDA material is considered as an isotropic linear model. In this regard, several authors have studied the stress–strain relationship for this type of material; some notable studies are those by Jeremic et al. [54], in which a cross-anisotropic elastic model is proposed, and by Meles et al. [55], in which several nonlinear relationships between stress and strain are proposed. Since these models seem more suitable to simulate the behaviour of these materials, this paper studied the influence of the constitutive model of the TDA material on the level of vibration transmission. Other authors, such as Medina et al. [33] have applied these constitutive models in cases of lightweight tunnel backfills, in which this material was used to reduce strains and stresses in the tunnel lining.

For the nonlinear isotropic model, the following stress–strain relationship was used: (6)$$\varepsilon =\frac{\sigma }{a+b\sigma }$$
where ε is the axial strain (%), σ the axial stress (kPa), and a and b are parameters that need to be adjusted. In this case, the values deduced by Medina et al. [33] were used and are shown in Table 5. In the case of the linear anisotropic model, the adopted values were also taken from [33] and are also shown in Table 5.

The results obtained with these constitutive models are compared in the frequency domain in Figure 18. In this figure, maximum differences of 3 dB can be seen at the points closest to the barrier (at 15 and 22.5 m) in the 20 to 40 Hz frequency range. The differences become almost imperceptible starting at the point located at 30 m. The reason for these small differences is that the nonlinear model works in a very small range of E values, since the level of strain induced in the barrier is very low; meanwhile, the anisotropic model considers two very low moduli of elasticity with a minor difference between them. Additionally, computation times increase by 25% for non-linear and linear anisotropic models on an Intel Core i7-4960X 64-bit computer, with 12 × 3.8 GHz processors, and 64 GB of RAM. Thus, it can be concluded that the constitutive model has little influence on the level of vibration reduction, and therefore, there is no reasonable justification for not using the simplest model, which is the linear isotropic model.

### 3.5. Comparison with Others Backfill Materials

The comparison of the results obtained with other common backfills for pile barriers was deemed of interest. In this case, concrete and steel tubular piles were considered. The mechanical properties of the concrete were: ρ = 2500 kg/m^3^, E = 30,000 MPa, ν = 0.2, ξ = 1%. The steel tubular piles were modelled using 1-cm-thick shell elements and the following mechanical properties: ρ = 7850 kg/m^3^, E = 200,000 MPa, ν = 0.3, ξ = 1%.

Figure 19 compares the results obtained in terms of IL and at different distances from the axis of the track. As shown, the most effective backfill was concrete, which reaches IL values of up to 27 dB, a very significant value. For steel tubular piles, the reduction is also relevant, reaching IL values of 25 dB. The piles with TDA backfill caused less reduction than that with concrete or with the steel tubular pile, although it offers interesting IL values that reach 10 dB in the 25–60 Hz frequency range, as shown previously. If the concrete and TDA backfills are compared, it is observed that the trend of the IL curve is similar but with different values. This implies that the key to vibration reduction is stiffness contrast, thus also confirming here the results of Thompson et al. [10]. The results also continue to show that increasing the distance to the barrier considerably reduces the mitigation effect.

Given these results, it seems obvious that backfills of very stiff materials (such as concrete) cause a very significant reduction in vibrations. Backfills made of very soft materials (such as TDA) cause a smaller but considerable reduction and can be an alternative to more common materials such as concrete. The reason the concrete and tubular steel piles outperform the TDA piles is related to the high bending stiffness of those materials, compared to the bending stiffness of TDA pile, which is very small. As shown [12], the transmission of plane waves in the ground with a wavelength smaller than the longitudinal bending wavelength of the barrier is impeded. In this way, the potential mitigation is more relevant for medium–high frequencies than for low frequencies. Then, the wave reflexion mechanism is more important for piles made with stiff materials than for those made with soft materials due to its high bending stiffness. Moreover, it is worth to indicate that in this case, the stiffness contrast between the concrete and the ground is more pronounced than that between the ground and the TDA, which may be one of the reasons for a greater reduction in the case of concrete piles. In addition, pile barriers with stiff backfills (concrete and steel tubular pile) cause wave reflexion, which is a mitigation mechanism that does not occur in TDA piles. Therefore, it is noteworthy that pile barriers made with concrete and tubular steel, whose damping is small (1%), cause a considerably greater reduction of vibrations than pile barriers with TDA (damping of 20%). In this sense, it could be deduced that damping is not the decisive parameter for the reduction of vibrations when pile barriers are treated. In these cases, the stiffness contrast is the most important parameter, and the reflexion mechanism is the more relevant. Obviously, a higher damping will cause a greater reduction of vibrations.

In economic and environmental sustainability terms, it must be taken into account that concrete and steel tubular piles have a high economic and environmental cost, while TDA is much less expensive and implies one of the most common measures in the fight against climate change: reuse and recycling of polluting materials. The comparison of the concrete and steel tubular piles clearly shows that concrete piles are more effective in terms of reducing vibrations as well as being a more economical solution than steel tubular piles. 

## 4. Conclusions

This paper analyses for the first time the efficiency of pile barriers filled exclusively with TDA material for the reduction of vibrations from rail traffic. This was done using a 3D finite-difference numerical model formulated in the time domain that was previously validated with data from a real case. The main conclusions are the following:-Pile barriers backfilled with TDA material are an effective measure for the reduction of railway vibrations, with IL values of up to 11 dB and low economic cost and are a very interesting measure in terms of environmental sustainability since the material originates from a highly polluting product that is recycled. The mitigation is caused by the damping of TDA and contrast of stiffness between TDA material and ground.-TDA pile wave barriers must be tangential in order to be effective since spacing the piles causes efficiency losses and the reduction effect is practically negligible given that the pile alone does not have vibration modes.-Regarding the effect of the constitutive model of the TDA material on the reduction of vibrations, it has been shown here that there is hardly any effect when the linear elastic model is compared with a hyperbolic model and with an anisotropic model.-The comparison with other backfills, such as concrete or steel tubular piles, shows that the TDA pile is a less effective measure for vibration reduction, although in terms of economic cost it is a very competitive solution, and it is also a very environmentally friendly solution.-The depth of the TDA barriers is closely related to the wavelength of the Rayleigh wave in the ground. In this regard, it should be noted that when a stiff soil layer is reached, the effect of lengthening or deepening the barrier under such a layer is very small, although in this sense, there are no general rules and each specific case should be carefully studied.

This research should be considered as a first study on the use of TDA material for the reduction of vibrations induced by rail traffic. Its real application would be the next stage, in which the mitigation effects could be validated more accurately by comparing real measurements with numerical results. In addition, further numerical studies could be carried out to evaluate its efficiency in other geotechnical and geometric conditions.

## Figures and Tables

**Figure 1 ijerph-17-09191-f001:**
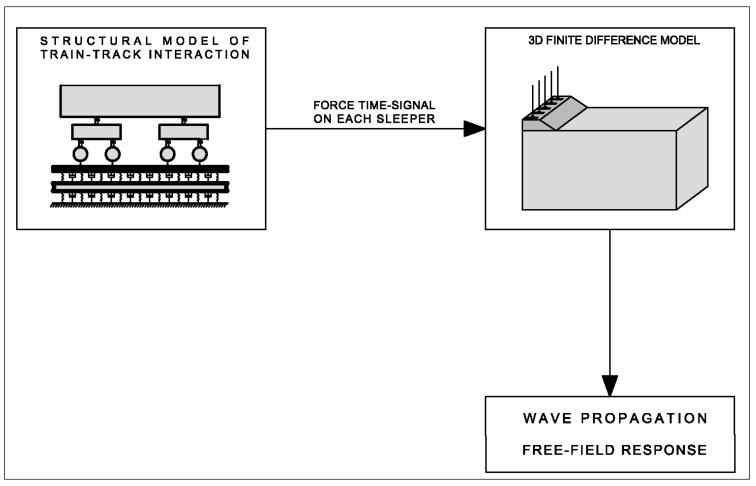
Computational scheme.

**Figure 2 ijerph-17-09191-f002:**
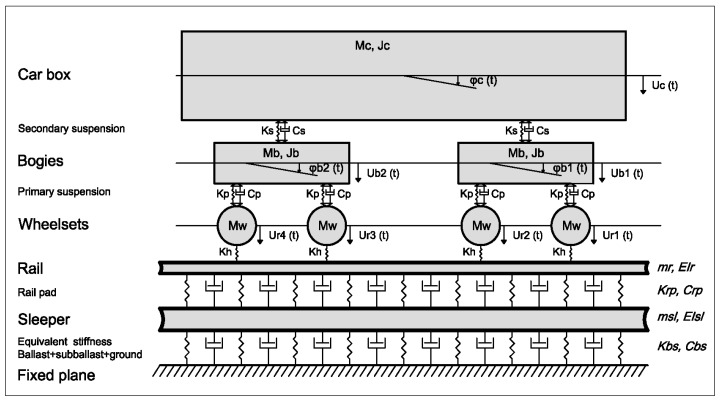
Train–track interaction model.

**Figure 3 ijerph-17-09191-f003:**
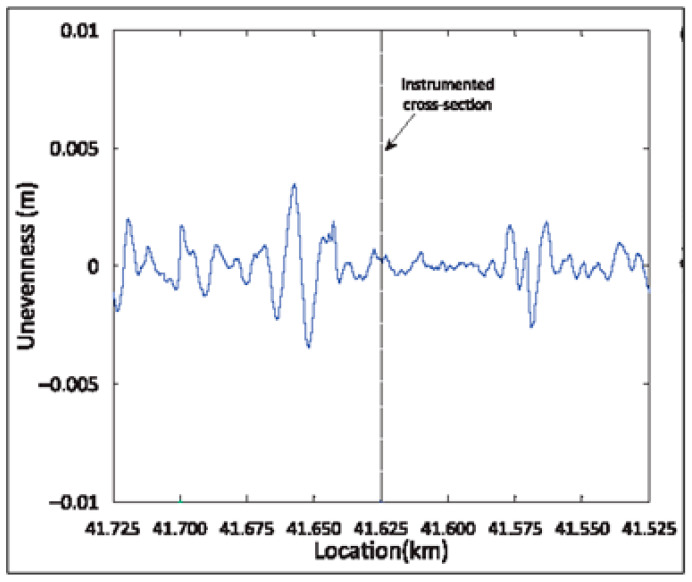
Unevenness rail profile.

**Figure 4 ijerph-17-09191-f004:**
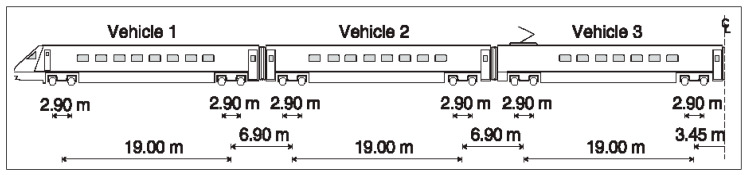
Alfa Pendular geometry.

**Figure 5 ijerph-17-09191-f005:**
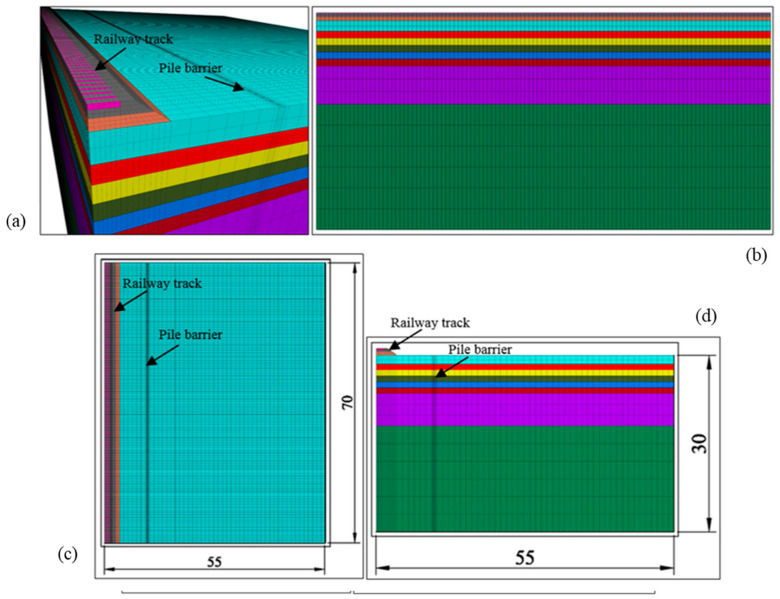
Numerical model for experimental validation (in metres): (**a**) perspective, (**b**) longitudinal section, (**c**) plan view, (**d**) cross section.

**Figure 6 ijerph-17-09191-f006:**
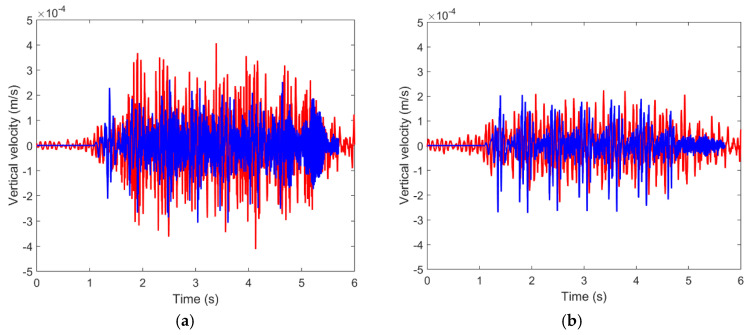
Time record of the vertical velocity at different distances: (**a**) 7 m; (**b**) 15 m; (**c**) 22.5 m; (**d**) 30 m; (**e**) 37.5 m; (**f**) 45 m; (red line: measured; blue line: numerical).

**Figure 7 ijerph-17-09191-f007:**
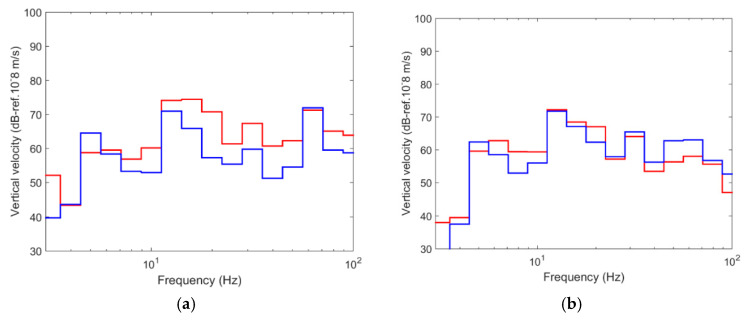
One-third octave spectra of the vertical velocity at distinct distances: (**a**) 7 m; (**b**) 15 m; (**c**) 22.5 m; (**d**) 30 m; (**e**) 37.5 m; (**f**) 45 m; (red line: measured; blue line: computed with FLAC3D).

**Figure 8 ijerph-17-09191-f008:**
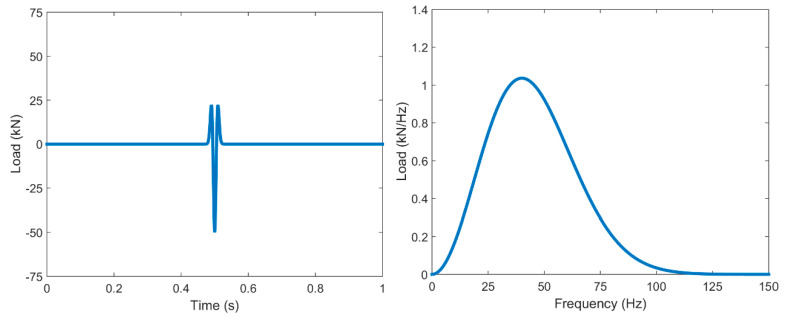
Stationary point load (left: time domain; right: frequency domain).

**Figure 9 ijerph-17-09191-f009:**
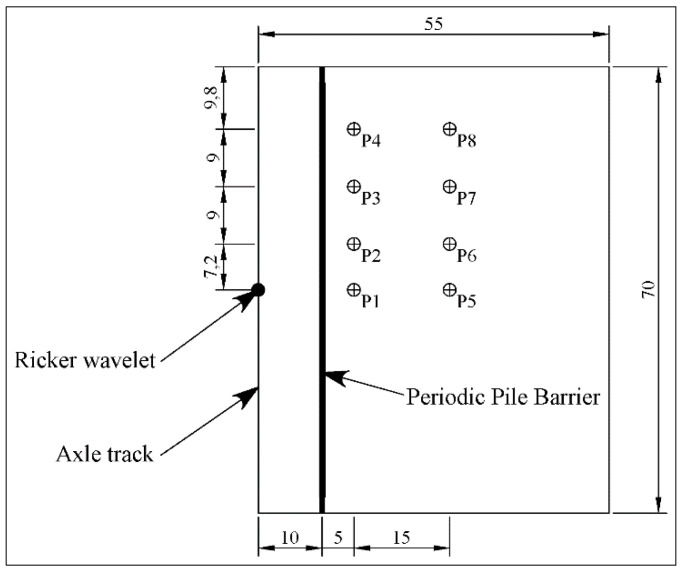
Schematic plan with reference points for stationary point load (in metres).

**Figure 10 ijerph-17-09191-f010:**
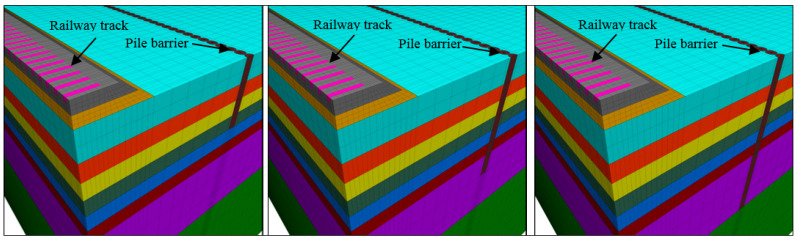
Numerical model for different barrier lengths: l = 4 m (**left**); l = 8 m (**middle**); l = 12 m (**right**).

**Figure 11 ijerph-17-09191-f011:**
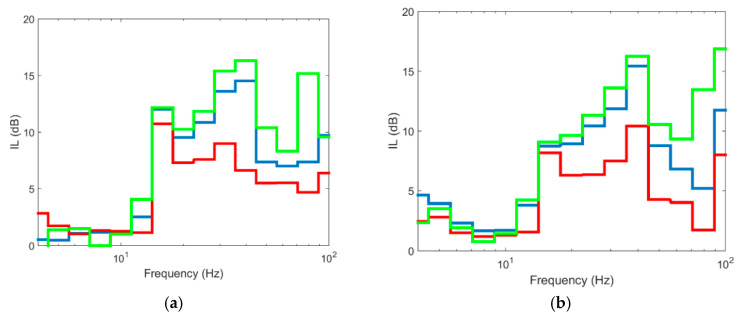
Insertion loss at different points in section to 15 m: (**a**) P1; (**b**) P2; (**c**) P3; (**d**) P4 (red line: pile length 4 m; blue line: pile length 8 m; green line: pile length 12 m).

**Figure 12 ijerph-17-09191-f012:**
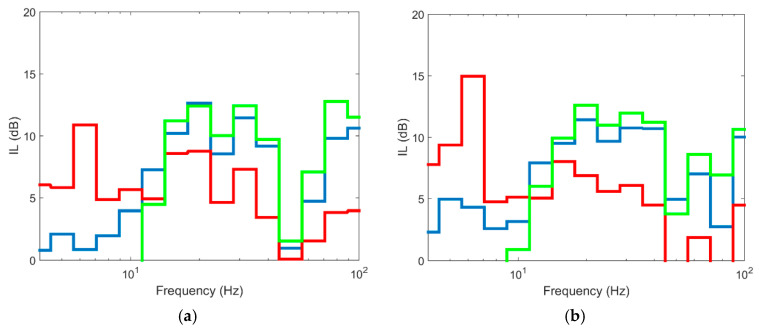
Insertion loss at different points in section to 30 m: (**a**) P5; (**b**) P6; (**c**) P7; (**d**) P8 (red line: pile length 4 m; blue line: pile length 8 m; green line: pile length 12 m).

**Figure 13 ijerph-17-09191-f013:**
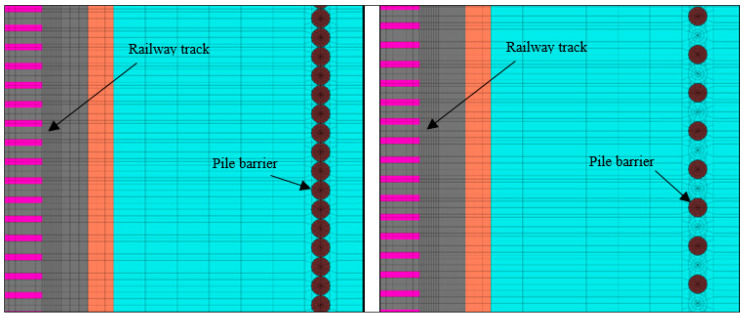
Numerical model for pile spacing: 1Φ (**left**); 2Φ (**right**).

**Figure 14 ijerph-17-09191-f014:**
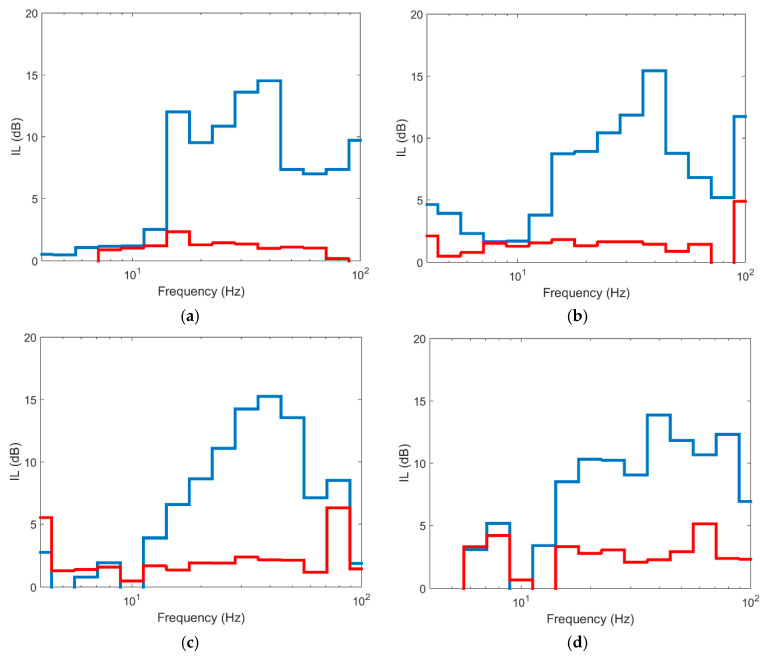
Insertion loss at different points in section to 15 m: (**a**) P1; (**b**) P2; (**c**) P3; (**d**) P4 (red line: pile spacing 2D; blue line: pile spacing 1D).

**Figure 15 ijerph-17-09191-f015:**
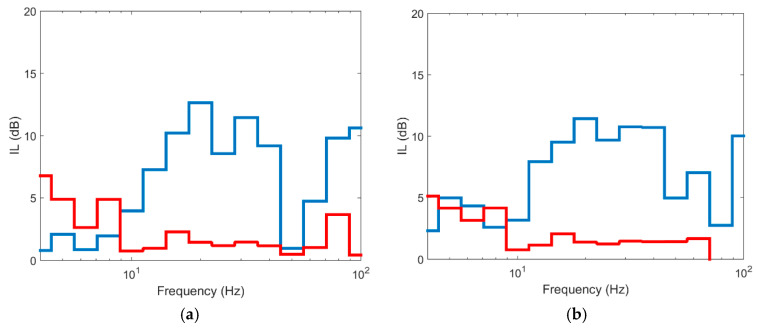
Insertion loss at different points in section to 30 m: (**a**) P5; (**b**) P6; (**c**) P7; (**d**) P8 (red line: pile spacing 2D; blue line: pile spacing 1D).

**Figure 16 ijerph-17-09191-f016:**
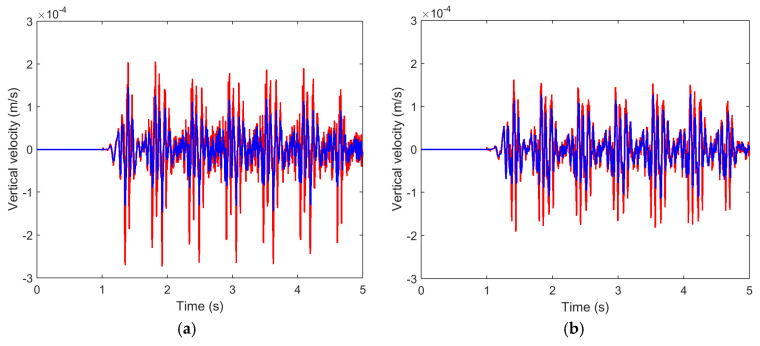
Time record of the vertical velocity at different distances: (**a**) 15 m; (**b**) 22.5 m; (**c**) 30 m; (**d**) 45 m; (red line: computed without piles; blue line: computed with tyre-derived-aggregate piles).

**Figure 17 ijerph-17-09191-f017:**
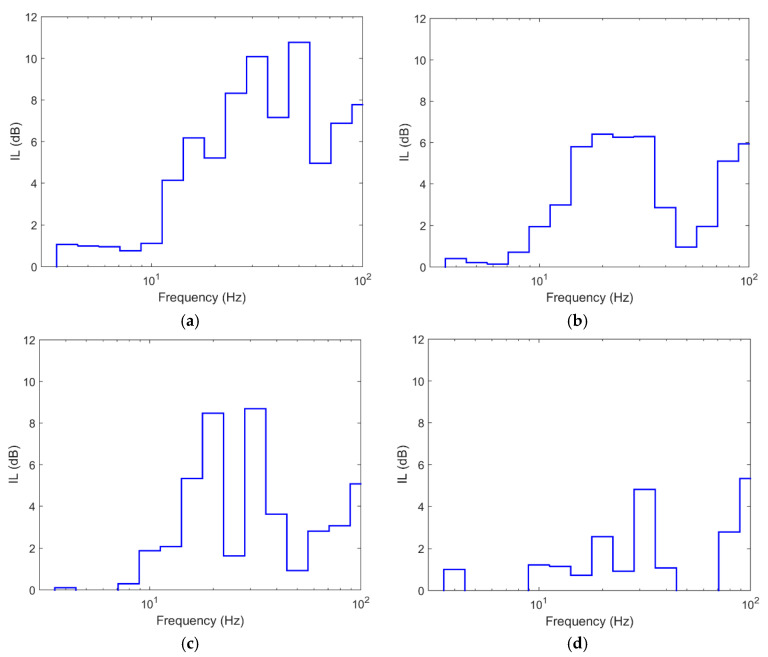
Insertion loss at different distances: (**a**) 15 m; (**b**) 22.5 m; (**c**) 30 m; (**d**) 45 m.

**Figure 18 ijerph-17-09191-f018:**
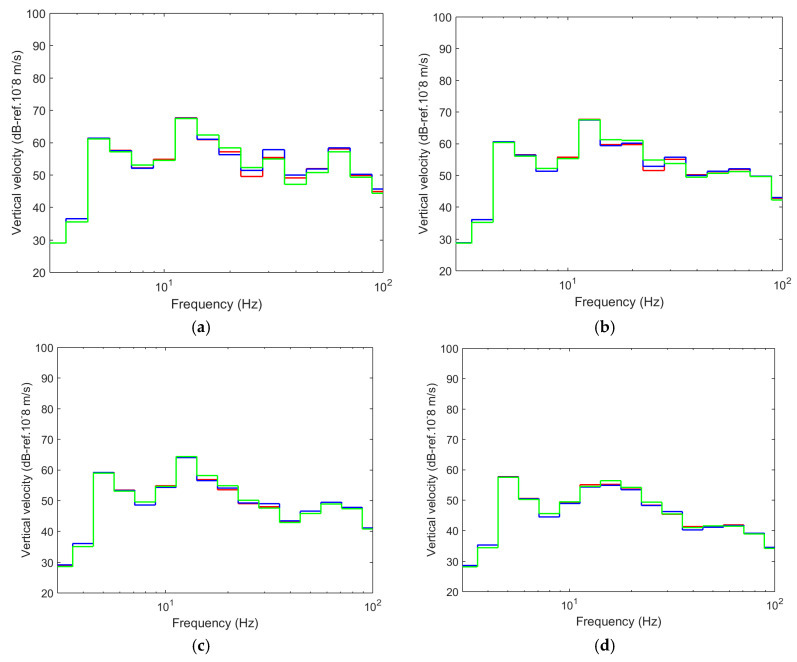
One-third octave spectra of the vertical velocity at distinct distances: (**a**) 15 m; (**b**) 22.5 m; (**c**) 30 m; (**d**) 45 m; (red line: linear isotropic; blue line: nonlinear isotropic; green line: linear anisotropic).

**Figure 19 ijerph-17-09191-f019:**
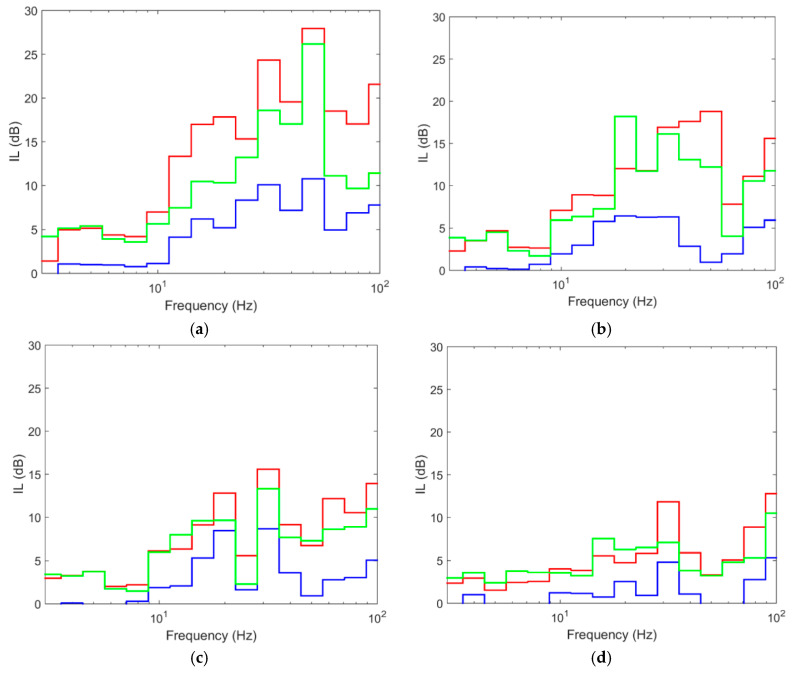
Insertion loss at different distances: (**a**) 15 m; (**b**) 22.5 m; (**c**) 30 m; (**d**) 45 m; (red line: computed with concrete piles; blue line: computed with tyre-derived-aggregate piles; green line: computed with steel tubular piles).

**Table 1 ijerph-17-09191-t001:** Ground elastic properties.

	Thickness (m)	ρ (kg/m^3^)	E (kN/m^2^)	ν	ξ (%)	Rayleigh Coefficients	
						α (s^−1^)	β (s)
Soil layer 1	1.5	1900	110.8 × 10^3^	0.48	3	5.65	0.000159
Soil layer 2	1.0	1900	95.8 × 10^3^	0.49	3	5.65	0.000159
Soil layer 3	1.0	1900	163.7 × 10^3^	0.49	3	5.65	0.000159
Soil layer 4	1.0	1900	119.5 × 10^3^	0.49	3	5.65	0.000159
Soil layer 5	1.0	1900	145.4 × 10^3^	0.49	3	5.65	0.000159
Soil layer 6	1.0	1900	226.6 × 10^3^	0.49	3	5.65	0.000159
Soil layer 7	5.5	1900	339.0 × 10^3^	0.48	3	5.65	0.000159
Soil layer 8	18.0	1900	539.6 × 10^3^	0.47	3	5.65	0.000159

**Table 2 ijerph-17-09191-t002:** Track elastic properties.

	Thickness (m)	ρ (kg/m^3^)	E (kN/m^2^)	ν	ξ (%)	Rayleigh Coefficients	
						α (s^−1^)	β (s)
Sleeper	0.22	2500	30 × 10^6^	0.20	1	1.88	0.000053
Ballast	0.35	1600	97 × 10^3^	0.12	6	11.30	0.000318
Sub-ballast	0.55	1900	212 × 10^3^	0.20	4	7.53	0.000212

**Table 3 ijerph-17-09191-t003:** Tyre-derived-aggregate (TDA) material’s elastic properties.

Tyre-Derived Aggregate	ρ (kg/m^3^)	E (kN/m^2^)	ν	ξ (%)	Rayleigh Coefficients	
					α (s^−1^)	β (s)
	640	630	0.20	20	37.68	0.00106

**Table 4 ijerph-17-09191-t004:** Gradation of the TDA material [33].

Maximum Size (mm)	Percentage Passing (%)
450	100
300	90
200	75
75	50
38	25
4.75	1

**Table 5 ijerph-17-09191-t005:** Mechanical properties adopted for the TDA layer.

Model	γ (kN/m^3^)	E (kPa)	ν	a	b	E_11_ (kPa)	E_33_ (kPa)	ν_12_ (=ν_31_)	ν_13_
Linear isotropic	6.4	630	0.20	-----	-----	-----	-----	-----	-----
Nonlinear isotropic	6.4	-----	0.20	248	2.65	-----	-----	-----	-----
Linear anisotropic	6.4	-----	-----	-----	-----	946	630	0.11	0.37

## Data Availability

Some or all data, models, or code that support the findings of this study are available from the corresponding author upon reasonable request (e.g., soil characterization, rail unevenness, train characteristics).

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
