# Peer review of "Use of Tyre-Derived Aggregate as Backfill Material for Wave Barriers to Mitigate Railway-Induced Ground Vibrations"

_ijerph, 2020, doi:10.3390/ijerph17249191_

Round 1
Reviewer 1 Report
This study focuses on the effect of using tire rubber as coarse aggregate in pile barrier on the mitigation of railway-induced ground vibration. The research topic is interesting. Overall, the paper is easy to follow. I have the following questions and suggestions:
(1) More detailed information about the pile barrier should be introduced:
What was the size of the granular tire rubber?
What were the dimensions of the pile barrier?
What was the layout the pile barrier in the ground?
How did you construct the pile barrier?
(2) More detailed information about the finite element model should be introduced:
How did you model the train, track, and ground?
How did you model their interface and interaction?
How many degrees of freedom did you consider for the train?
What were the parameters for the mechanical properties of the train and track?
You may refer to the following paper: Gou, H., Yang, L., Mo, Z., et al., 2019. Effect of Long-Term Bridge Deformations on Safe Operation of High-Speed Railway and Vibration of Vehicle–Bridge Coupled System. International Journal of Structural Stability and Dynamics, 19(09), p.1950111.
(3) How did you validate the finite element model before it is used for further analysis?
(4) What is "IL"? Please clearly define every acronym. You are suggested to minimize the use of acronyms.
(5) How did you model the interaction between tire rubber and soil in your finite element model? Did you consider the interaction between different piece of tire rubber?
(6) Please further discuss the damping of different types of materials, such as rubber, concrete, and steel, which are mentioned in this paper. Is the mitigation of vibration purely related to damping? The mechanism of the mitigation should be elaborated.
Author Response
COVER LETTER
Title: Use of tire‐derived aggregate as backfill material for pile wave barriers to mitigate railway‐induced ground vibrations
Authors: Fernández-Ruiz, Jesús; Medina Rodríguez, Luis; Alves Costa, Pedro
- Initial Considerations
1.1 We would like to start by expressing our gratitude to all the reviewers for the relevant comments addressed in their reports about the manuscript referenced above. These comments and suggestions helped us to improve the manuscript in specific aspects.
1.2 This letter serves as a guide in the identification of all the changes introduced into the reviewed version of the manuscript, in order to respond to your pertinent criticisms and suggestions. The most relevant changes in the manuscript are identified by means of underlined text.
1.3 Prior to explicitly specifying such revisions, we would like to remark here that all the questions presented in the reviewers’ reports were, as far as possible, included and responded too.
- Specific Answers to the Reviewers’ Comments
Concerning the pertinent remarks raised by the reviewer (italic text), the following changes were introduced into the manuscript, or alternatively the following clarification comments are made:
GENERAL COMMENT
This study focuses on the effect of using tire rubber as coarse aggregate in pile barrier on the mitigation of railway-induced ground vibration. The research topic is interesting. Overall, the paper is easy to follow. I have the following questions and suggestions:
The authors thank the general comment of the reviewer and would like to state that the suggestions presented by the reviewer were appreciated and considered in the improving of the paper.
SPECIFIC COMMENTS
(1) More detailed information about the pile barrier should be introduced:
What was the size of the granular tire rubber?
In line 185 is added the following text:
Concerning to the TDA material it is made from used conventional commercial tires. Table 3 shows its the mechanical properties, considering a linear elastic model [33]. These properties correspond to the gradation of the TDA material shown in table 4……………………
Table 4. Gradation of the TDA material [33]
|
Maximum size (mm) |
Percentage passing (%) |
|
450 |
100 |
|
300 |
90 |
|
200 |
75 |
|
75 |
50 |
|
38 |
25 |
|
4.75 |
1 |
What were the dimensions of the pile barrier?
In line 266 is added the following text:
In all the numerical models made in this paper, the diameter of the piles has been 0.6 m, which may be considered as the smallest dimension in practice. A smaller diameter could be problematic during the construction and filling processes. Only this diameter has been considered in this paper because the barrier width seems not to have a relevant influence on results [10]. Moreover, the length of pile barrier has been considered as 4, 8 and 12 meters.
What was the layout the pile barrier in the ground?
The barrier crosses several layers of ground which are shown in figure 10.
Figure 10. Numerical model for different barrier lengths: l = 4 m (left); l = 8 m (middle); l = 12 m (right)
How did you construct the pile barrier?
The pile barrier was not built. This is a numerical study on its possible effects. Its considerations for being built are shown on lines 67-74.
(2) More detailed information about the finite element model should be introduced:
How did you model the train, track, and ground?
How did you model their interface and interaction?
How many degrees of freedom did you consider for the train?
These three interesting questions are introduced in line 106, as follows:
In this model, the train is completely considered (including all the masses) through rigid body dynamics method with 10 degrees of freedom. The track is simulated as combination of masses and linear springs-dashpots and the ballast, subballast and ground are replaced by a linear equivalent spring-dashpot. The rail and the sleepers are modelled as infinitive Euler-Bernoulli beams and the railpad as a linear spring-dashpot. The train and track are interrelated through the wheel-rail Hertz contact. The resultant forces over the sleepers are computed and later applied in the numerical model, as summarized in fig. 1. Similar models are common in the technical literature [44], among others.
What were the parameters for the mechanical properties of the train and track?
You may refer to the following paper: Gou, H., Yang, L., Mo, Z., et al., 2019. Effect of Long-Term Bridge Deformations on Safe Operation of High-Speed Railway and Vibration of Vehicle–Bridge Coupled System. International Journal of Structural Stability and Dynamics, 19(09), p.1950111.
In line 181 is added the following text:
Related to the ground and track elastic properties shown in Table 1 and 2, it should be noted that all values were obtained from dynamic tests (cross-hole, receptance tests, etc.).
The above reference is introduced as [44] in line 113.
(3) How did you validate the finite element model before it is used for further analysis?
In line 161 is added the following text:
The numerical scheme used was experimentally validated on the Lisbon-Porto line, located near the town of Carregado (Portugal) by comparison between vertical vibrations at different points on the ground. These vibrations were induced by railway traffic in common traffic operations.
(4) What is "IL"? Please clearly define every acronym. You are suggested to minimize the use of acronyms.
In line 307 is added the following text:
The IL is defined as follow:
(5)
where vref is the computed vertical velocity in the numerical model without pile barriers and v is the computed vertical velocity in the numerical model with pile barriers.
(5) How did you model the interaction between tire rubber and soil in your finite element model? Did you consider the interaction between different piece of tire rubber?
In line 194 is added the following text:
The shared node method is considered for the TDA piles-ground interaction because the relative movement between them is very small, so the friction between piles and soils is neglected in this research. In the same way, the interaction between different pieces of tire rubber has also been neglected as the TDA is modelled as a continuous medium.
(6) Please further discuss the damping of different types of materials, such as rubber, concrete, and steel, which are mentioned in this paper. Is the mitigation of vibration purely related to damping? The mechanism of the mitigation should be elaborated.
In line 480 is added the following text:
Given these results, it seems obvious that backfills of very stiff materials (such as concrete) cause a very significant reduction in vibrations. Backfills made of very soft materials (such as TDA) cause a smaller but considerable reduction and can be an alternative to more common materials such as concrete. The reason why the concrete and tubular steel piles outperform the TDA piles is related to the high bending stiffness of those, being the bending stiffness of TDA pile very small. As shown [12], the transmission of plane waves in the ground with a wavelength smaller than the longitudinal bending wavelength of the barrier is impeded. In this way, the potential mitigation is more relevant for medium-high frequencies than for low frequencies. Then, the wave reflexion mechanism is more important for piles made with stiff materials than with soft materials due to its high bending stiffness. Moreover, it is worth to indicate that in this case, the stiffness contrast between the concrete and the ground is more pronounced than between the ground and the TDA, which may be one of the reasons for a greater reduction in the case of concrete piles. In addition, the pile barrier with stiff backfills (concrete and steel tubular pile) cause wave reflex-ion, which is a mitigation mechanism that does not occur in TDA piles. Therefore, it is noteworthy that pile barriers made with concrete and tubular steel, whose damping are small (1%), cause a considerably greater reduction of vibrations than pile barriers with TDA (damping of 20%). In this sense, it could be deduced that damping is not the decisive parameter for the reduction of vibrations when pile barriers are treated. In these cases, the stiffness contrast is the most important parameter and the reflexion mechanism is the more relevant. Obviously, a higher damping will cause a greater reduction of vibrations.

Reviewer 2 Report
This paper analyses the efficiency of pile barriers filled exclusively with TDA material for the reduction of vibrations from rail traffic. A 3D finite-difference numerical model formulated in the time domain is employed to simulate the vibration from railway. It could be considered for publication after stressing the follow issues.
- It’s better to give a photograph or an imaginary photograph of the pile barriers arrangement on railway system. This will help readers unfamiliar to this area to see where these piles locate, thus understanding your research rapidly.
- In Page 31, the authors said “it seems obvious that backfills of very stiff materials (such as concrete) cause a very significant reduction in vibrations, while backfills made of very soft materials (such as TDA) cause a smaller but considerable reduction and can be an alternative to more common materials such as concrete.” It’s better to give more explanation about the possible mechanisms.
- In Fig.5, 10 and 13, it’s better to mark the name of different parts, such as piles, and railways.
Author Response
COVER LETTER
Title: Use of tire‐derived aggregate as backfill material for pile wave barriers to mitigate railway‐induced ground vibrations
Authors: Fernández-Ruiz, Jesús; Medina Rodríguez, Luis; Alves Costa, Pedro
- Initial Considerations
1.1 We would like to start by expressing our gratitude to all the reviewers for the relevant comments addressed in their reports about the manuscript referenced above. These comments and suggestions helped us to improve the manuscript in specific aspects.
1.2 This letter serves as a guide in the identification of all the changes introduced into the reviewed version of the manuscript, in order to respond to your pertinent criticisms and suggestions. The most relevant changes in the manuscript are identified by means of underlined text.
1.3 Prior to explicitly specifying such revisions, we would like to remark here that all the questions presented in the reviewers’ reports were, as far as possible, included and responded too.
- Specific Answers to the Reviewers’ Comments
Reviewer 2
GENERAL COMMENT
This paper analyses the efficiency of pile barriers filled exclusively with TDA material for the reduction of vibrations from rail traffic. A 3D finite-difference numerical model formulated in the time domain is employed to simulate the vibration from railway. It could be considered for publication after stressing the follow issues.
The authors thank the general comment of the reviewer and would like to state that the suggestions presented by the reviewer were appreciated and considered in the improving of the paper.
SPECIFIC COMMENTS
- It’s better to give a photograph or an imaginary photograph of the pile barriers arrangement on railway system. This will help readers unfamiliar to this area to see where these piles locate, thus understanding your research rapidly.
The authors have improved some figures and they believe that now it is clearer this research. The figures are:
Figure 5. Numerical model for experimental validation (in metres): a) perspective, b) longitudinal section, c) plan view, d) cross section
Figure 9. Schematic plan with reference points for stationary point load (in metres)
Figure 10. Numerical model for different barrier lengths: l = 4 m (left); l = 8 m (middle); l = 12 m (right)
Figure 13. Numerical model for pile spacing: 1Φ (left); 2Φ (right)
- In Page 31, the authors said “it seems obvious that backfills of very stiff materials (such as concrete) cause a very significant reduction in vibrations, while backfills made of very soft materials (such as TDA) cause a smaller but considerable reduction and can be an alternative to more common materials such as concrete.” It’s better to give more explanation about the possible mechanisms.
In line 480 is added the following text:
Given these results, it seems obvious that backfills of very stiff materials (such as concrete) cause a very significant reduction in vibrations. Backfills made of very soft materials (such as TDA) cause a smaller but considerable reduction and can be an alternative to more common materials such as concrete. The reason why the concrete and tubular steel piles outperform the TDA piles is related to the high bending stiffness of those, being the bending stiffness of TDA pile very small. As shown [12], the transmission of plane waves in the ground with a wavelength smaller than the longitudinal bending wavelength of the barrier is impeded. In this way, the potential mitigation is more relevant for medium-high frequencies than for low frequencies. Then, the wave reflexion mechanism is more important for piles made with stiff materials than with soft materials due to its high bending stiffness. Moreover, it is worth to indicate that in this case, the stiffness contrast between the concrete and the ground is more pronounced than between the ground and the TDA, which may be one of the reasons for a greater reduction in the case of concrete piles. In addition, the pile barrier with stiff backfills (concrete and steel tubular pile) cause wave reflex-ion, which is a mitigation mechanism that does not occur in TDA piles. Therefore, it is noteworthy that pile barriers made with concrete and tubular steel, whose damping are small (1%), cause a considerably greater reduction of vibrations than pile barriers with TDA (damping of 20%). In this sense, it could be deduced that damping is not the decisive parameter for the reduction of vibrations when pile barriers are treated. In these cases, the stiffness contrast is the most important parameter and the reflexion mechanism is the more relevant. Obviously, a higher damping will cause a greater reduction of vibrations.
- In Fig.5, 10 and 13, it’s better to mark the name of different parts, such as piles, and railways.
These names have been included in figures 5, 10 and 13. Please see first response to this reviewer.

Reviewer 3 Report
Comments to the authors:
- In abstract, some quantitative final results should be presented.
- In “Material and methods”, it is not clear which elastomeric material was used ? A clear description of material including its overall physical and mechanical properties should be present in this section.
- The lifetime and durability of the used compound should be discussed.
- The weather conditions of the tests should be presented. Also, it is interesting to consider the effect of environment’s temperature and humidity on the material performance, as the elastomers behave different in different weathers.
Best Regards,
Reviewer
Author Response
COVER LETTER
Title: Use of tire‐derived aggregate as backfill material for pile wave barriers to mitigate railway‐induced ground vibrations
Authors: Fernández-Ruiz, Jesús; Medina Rodríguez, Luis; Alves Costa, Pedro
- Initial Considerations
1.1 We would like to start by expressing our gratitude to all the reviewers for the relevant comments addressed in their reports about the manuscript referenced above. These comments and suggestions helped us to improve the manuscript in specific aspects.
1.2 This letter serves as a guide in the identification of all the changes introduced into the reviewed version of the manuscript, in order to respond to your pertinent criticisms and suggestions. The most relevant changes in the manuscript are identified by means of underlined text.
1.3 Prior to explicitly specifying such revisions, we would like to remark here that all the questions presented in the reviewers’ reports were, as far as possible, included and responded too.
- Specific Answers to the Reviewers’ Comments
Reviewer 3
SPECIFIC COMMENTS
- In abstract, some quantitative final results should be presented.
In line 21 the following text has been added:
The numerical results show Insertion Loss (IL) values of up to 11 dB for a depth closed to the wavelength of Rayleigh wave. Finally, this solution is compared with more common backfills, such as concrete and steel tubular piles, showing that the TDA pile is a less effective measure although from an environmental and engineering point of view it is a very competitive solution.
- In “Material and methods”, it is not clear which elastomeric material was used? A clear description of material including its overall physical and mechanical properties should be present in this section.
In line 185 the following text has been added:
Concerning to the TDA material it is made from used conventional commercial tires. Table 3 shows its the mechanical properties, considering a linear elastic model [33]. These properties correspond to the gradation of the TDA material shown in table 4 and were obtained under laboratory conditions. However, there are some available field tests with which similar results were obtained [50]. Although there is still no experimental evidence in civil engineering applications, it is well known that the durability and lifetime of this material is very long, beyond the design life of the constructions in which it is used. The TDA material is impermeable, inert, and its mechanical behaviour depends in a very little manner on the environmental conditions where it is employed.
Moreover, the table 3 has been moved to this section (line 208) and a new table (table 4) has been introduced:
Table 4. Gradation of the TDA material [33]
|
Maximum size (mm) |
Percentage passing (%) |
|
450 |
100 |
|
300 |
90 |
|
200 |
75 |
|
75 |
50 |
|
38 |
25 |
|
4.75 |
1 |
- The lifetime and durability of the used compound should be discussed.
- The weather conditions of the tests should be presented. Also, it is interesting to consider the effect of environment’s temperature and humidity on the material performance, as the elastomers behave different in different weathers.
These two questions are discussed in line 185:
Concerning to the TDA material it is made from used conventional commercial tires. Table 3 shows its the mechanical properties, considering a linear elastic model [33]. These properties correspond to the gradation of the TDA material shown in table 4 and were obtained under laboratory conditions. However, there are some available field tests and similar results were obtained [50]. Although there is still no experimental evidence in civil engineering applications, it is well known that the durability and lifetime of this material is very long, beyond the design life of the constructions in which it is used. The TDA material is impermeable, inert, and its mechanical behaviour depends in a very little manner on the environmental conditions where it is employed.

Reviewer 4 Report
Manuscript ID: ijerph‐1019196
Title of paper: Use of tire‐derived aggregate as backfill material for pile wave barriers to mitigate
railway‐induced ground vibrations
Comments and Suggestions for Authors
This manuscript includes an experimental research about the use of tire‐derived aggregate as backfill material for pile wave barriers to mitigate railway‐induced ground vibrations. First of all, in my opinion, the topic of the paper is interesting and novel.
Nevertheless, I have several suggestions in order to improve the manuscript before being published in IJERPH journal.
First of all, as general comment, the manuscript must be formatted following the template and the instructions for authors of MDPI journals.
Firstly, in relation to the introduction section, the state‐of‐art review included here is very good, with many references cited. However, in my opinion the aim of the research work included in the paper should be clearly explained at the end of this section. I suggest to include a final paragraph in with a more explicit description of the objectives and the innovation of this work.
The materials and methods section is generally adequate. Regarding the study case described in section 2.4, why has it been considered an Alfa‐Pendular passenger train? Maybe, it could be more suitable to consider a freight train, whose speed was lower and its load was higher, or even, an Intercidades train, with a higher speed than freight one, and with some elements like locomotive
5600 series with relatively heavy axle load.
In my opinion, the current section “3 Results of pile barriers of TDA” should be renamed as “3 Results and discussion”. The description of results and their discussion explained here is very good and many references have been cited for supporting the results obtained.
Regarding the conclusion section, I think that it is fine. I like the idea of using bullet points for highlighting the main findings of the manuscript, this makes clearer this section.
Finally, despite the comments, I want to encourage the authors for continuing working in this research topic, and I think that the manuscript could be accepted for publication in IJERPH journal after including the proposed changes.
Author Response
COVER LETTER
Title: Use of tire‐derived aggregate as backfill material for pile wave barriers to mitigate railway‐induced ground vibrations
Authors: Fernández-Ruiz, Jesús; Medina Rodríguez, Luis; Alves Costa, Pedro
- Initial Considerations
1.1 We would like to start by expressing our gratitude to all the reviewers for the relevant comments addressed in their reports about the manuscript referenced above. These comments and suggestions helped us to improve the manuscript in specific aspects.
1.2 This letter serves as a guide in the identification of all the changes introduced into the reviewed version of the manuscript, in order to respond to your pertinent criticisms and suggestions. The most relevant changes in the manuscript are identified by means of underlined text.
1.3 Prior to explicitly specifying such revisions, we would like to remark here that all the questions presented in the reviewers’ reports were, as far as possible, included and responded too.
- Specific Answers to the Reviewers’ Comments
Reviewer 4
GENERAL COMMENT
This manuscript includes an experimental research about the use of tire‐derived aggregate as backfill material for pile wave barriers to mitigate railway‐induced ground vibrations. First of all, in my opinion, the topic of the paper is interesting and novel.
Nevertheless, I have several suggestions in order to improve the manuscript before being published in IJERPH journal.
The authors thank the general comment of the reviewer and would like to state that the suggestions presented by the reviewer were appreciated and considered in the improving of the paper.
SPECIFIC COMMENTS
First of all, as general comment, the manuscript must be formatted following the template and the instructions for authors of MDPI journals.
The manuscript has been formatted following the template and the instructions for authors of MDPI journals.
Firstly, in relation to the introduction section, the state‐of‐art review included here is very good, with many references cited. However, in my opinion the aim of the research work included in the paper should be clearly explained at the end of this section. I suggest to include a final paragraph in with a more explicit description of the objectives and the innovation of this work.
This paragraph has been added in line 86, as follows:
Therefore, in this paper and for the first time are studied wave pile barriers made exclusively with TDA as a vibration mitigation measure for rail transportation systems. This is novel from an environmental and engineering point of view because a highly polluting waste can be reused as a backfill material for reduction of railway vibrations. Hence, the main aims of this study are to analyse the level of vibration reduction on the ground caused by this kind of pile barriers, studying both the effect of the depth and the pile spacing.
The materials and methods section is generally adequate. Regarding the study case described in section 2.4, why has it been considered an Alfa‐Pendular passenger train? Maybe, it could be more suitable to consider a freight train, whose speed was lower and its load was higher, or even, an Intercidades train, with a higher speed than freight one, and with some elements like locomotive 5600 series with relatively heavy axle load.
This interesting question has been added in line 169, as follows:
The Alfa-Pendular train has been considered because the problem associated with railway-induced vibrations on the ground is usually limited to passenger trains (without heavy axle loads) where its high speed causes high frequency vibrations, which are perceived far away from the railway track (tens of meters). On the contrary, the highest vibrations produced by freight trains (heavy axle loads) are limited to the railway track, since its low speed implies low frequency vibrations (< 10 Hz), which are not perceived at important distances of the railway track.
In my opinion, the current section “3 Results of pile barriers of TDA” should be renamed as “3 Results and discussion”. The description of results and their discussion explained here is very good and many references have been cited for supporting the results obtained.
This question has been considered and the section 3 has been renamed as “Results and discussion” (see line 258).
Regarding the conclusion section, I think that it is fine. I like the idea of using bullet points for highlighting the main findings of the manuscript, this makes clearer this section.
The authors thank this comment.
Finally, despite the comments, I want to encourage the authors for continuing working in this research topic, and I think that the manuscript could be accepted for publication in IJERPH journal after including the proposed changes.
The authors thank this comment.

Round 2
Reviewer 1 Report
The authors have addressed my comments.